# Bridging the Gap between Recognition-level Pre-training and Commonsensical Vision-language Tasks

**Yue Wan[*], Yueen Ma[*], Haoxuan You, Zhecan Wang, Shih-Fu Chang**
Columbia University
{yw3373, ym2745, hy2612, zw2627, sc250}@columbia.edu

## Abstract

Large-scale visual-linguistic pre-training aims to capture the generic representations from multimodal features, which are essential for downstream vision-language tasks. Existing methods mostly focus on learning the semantic connections between visual objects and linguistic content, which tend to be recognition-level information and may not be sufficient for commonsensical reasoning tasks like VCR. In this paper, we propose a novel commonsensical vision-language pre-training framework to bridge the gap. We first augment the conventional image-caption pre-training datasets with commonsense inferences from a visual-linguistic GPT-2. To pre-train models on image, caption and commonsense inferences together, we propose two new tasks: *masked commonsense modeling* (MCM) and *commonsense type prediction* (CTP). To reduce the shortcut effect between captions and commonsense inferences, we further introduce the *domain-wise adaptive masking* that dynamically adjusts the masking ratio. Experimental results on downstream tasks, VCR and VQA, show the improvement of our pre-training strategy over previous methods. Human evaluation also validates the relevance, informativeness, and diversity of the generated commonsense inferences. Overall, we demonstrate the potential of incorporating commonsense knowledge into the conventional recognition-level visual-linguistic pre-training.

## 1 Introduction

Vision-language multimodal tasks have received vast attention in the deep learning field in recent years. Tasks, like Visual Question Answering (VQA) (Antol et al., 2015; Goyal et al., 2017) and Visual Commonsense Reasoning (VCR) (Zellers et al., 2019), require different levels of multimodal reasoning ability to make task-specific decisions.

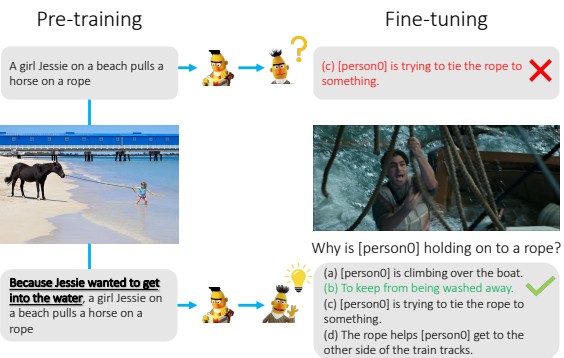

Figure 1: An example of our commonsensical visual-linguistic pre-training (bottom) compared against the conventional visual-linguistic pre-training (top). Commonsensical knowledge (e.g., the bold underlined text) is generated and learned by models during our commonsensical pre-training. Such knowledge becomes useful for downstream commonsense reasoning tasks: our model correctly answers the question while the conventional method is wrong.

Motivated by the advancement of pre-training in both computer vision (CV), such as backbone networks pre-trained on ImageNet (Deng et al., 2009), and natural language processing (NLP), such as BERT (Devlin et al., 2018) and GPT-2 (Radford et al., 2019), numerous visual-linguistic pre-training strategies were proposed to learn the generic feature representations for vision-language tasks. Most of them (Su et al., 2020; Lu et al., 2019a; Chen et al., 2020; Tan and Bansal, 2019; Gan et al., 2020) take advantage of large-scale image captioning datasets, such as Conceptual Captions (Sharma et al., 2018) and MSCOCO Captions (Lin et al., 2014). These pre-training tasks mostly focus on learning the modality alignments between regions-of-interest (RoIs) from images and words from captions by applying the visual-linguistic extensions of the *masked language modeling* (MLM) objective. There are also other multimodal objectives, such as *word-region alignment* (Lu et al., 2019a; Chen et al., 2020), *image-text matching* (Chen et al., 2020) and *scene graph prediction* (Yu

[*]These authors contributed equally. The majority of this work is finished during their master's degree at Columbia University.

| | Recognition-level | Commonsensical | |
|---|---|---|---|
| Type | Low-level Caption | Commonsense Inference | High-level Caption |
| Dataset | MSCOCO | VisualCOMET | Ours |
| Example | A girl Jessie on a beach pulls a horse on a rope | <intent> **get into the water** | Because Jessie wanted to **get into the water**, a girl Jessie on a beach pulls a horse on a rope. |

Table 1: Terminologies used in this paper, along with their corresponding datasets and examples. The bold text represents the commonsense inference and the underlined text represents template tokens for the commonsense type, <intent>. The example captions correspond to the left image in Figure 1.

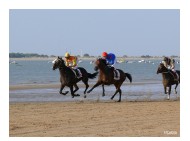

(a) Recognition-level VQA Example.
Q: What are the people racing?
A: Horses.

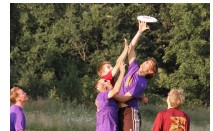

(b) Commonsensical VQA Example.
Q: Why are the men jumping?
A: To catch frisbee.

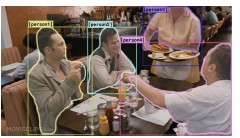

(c) VCR Example (Commonsensical).
Q: Why is [person4] pointing at [person1]?
A: He is telling [person3] that [person1] ordered the pancakes.

Figure 2: Recognition-level and commonsensical visual question answering examples from VQA and VCR.

et al., 2020).

Despite the variety of those proposed pre-training strategies, they mostly capture the recognition-level relationship between the two modalities, which might not be sufficient for vision-language tasks that require cognition-level reasoning abilities. Here, the term *cognition* is taken from VCR to represent reasoning abilities and is more advanced than *recognition*. In this work, we rephrase *cognition-level* as *commonsensical* to avoid confusion. As an example, being aware of the word "man" referring to the human-alike object in the image is insufficient to infer his future behavior. Su et al. (2020); Chen et al. (2020) also showed the similar discrepancy between recognition-level pre-training and commonsensical fine-tuning. Thus, the motivation of this work is to bridge the gap between the two learning stages for vision-language reasoning tasks.

Not to be confused with the term "commonsense" described in CommonsenseQA (Talmor et al., 2019), we approach it from a cognitive perspective and take the concept of "commonsense inference" proposed in VisualCOMET (Park et al., 2020) as the starting point. It introduced three specific types of commonsense knowledge, which are the possible incidents before or after the current event (i.e., temporal), and the potential intentions of the target subjects (i.e., intentional). Unfortunately, these information does not normally exist in conventional captions. Therefore, a natural question would be whether introducing additional commonsense knowledge in pre-training can further improve upon the downstream commonsensical

tasks.

To answer this question, we develop a novel commonsensical vision-language pre-training framework, which contains two main components: (1) Generating commonsense inferences for the conventional image-caption dataset; (2) Introducing suitable pre-training strategies for image, caption, and commonsense inference together.

As for commonsense inference generation, we fine-tune a visual-linguistic GPT-2 on Visual-COMET (Park et al., 2020) as our commonsense generator and infer the temporal and intentional commonsense for the image-caption pairs in MSCOCO dataset. We define the conventional captions such as MSCOCO captions as the *low-level* captions. We then combine the low-level captions with the commonsense inferences using pre-defined templates to get the *high-level* captions. The terminologies used in this paper are collected in Table 1 and examples are shown in Figure 2.

Given additional commonsense inferences besides the image and caption, the pre-training strategy is the key to bridge the recognition-level information and commonsense. In short, we replace the low-level captions used in most conventional pre-training methods with the high-level captions. We propose two tasks toward commonsense inferences: *masked commonsense modeling* (MCM) and *commonsense type prediction* (CTP). MCM requires the model to predict the commonsense inference masked by the *domain-wise adaptive masking* strategy. It dynamically adjusts the masking ratio based on the semantic similarity between commonsense inferences and captions, for the sake of avoiding ob-

vious shortcuts. In CTP, the type of commonsense among <intent>, <before> or <after> is predicted without knowing the template tokens, which forces the model to learn global relationships among commonsense, captions, and images.

Eventually, we take VCR and VQA as two downstream evaluation tasks to demonstrate the effectiveness of our framework. We further provide qualitative analysis and human evaluation to reveal the insights behind it.

Our main contributions in this paper are:

- We propose a novel commonsensical visual-linguistic pre-training framework for incorporating commonsense knowledge into the conventional image-caption pre-training;

- We fine-tune a visual-linguistic GPT-2 model as the commonsense generator that takes as input a low-level image-caption pair;

- We develop two commonsensical pre-training tasks—MCM and CTP, which encourages the model to internalize commonsensical reasoning ability;

- We conduct comprehensive comparison and ablation study to show that our pre-training framework leads to improvements of 1.43% on VCR and 1.26% on VQA. Moreover, a human evaluation is conducted to validate the quality of the generated commonsense inferences.

## 2 Related Work

### 2.1 Visual-linguistic Model

Vision and language models have been advancing rapidly and, with the introduction of Faster R-CNN (Ren et al., 2015) and Transformer-based models (Vaswani et al., 2017) (e.g., GPT (Radford et al., 2018, 2019; Brown et al., 2020) and BERT (Devlin et al., 2018)), many vision-language tasks are becoming easier to solve. The original BERT can be easily extended to vision-language multimodal settings by concatenating the visual features of regions-of-interest (RoIs) and linguistic features of word tokens. Multiple BERT variants were introduced to solve the *visual question answering* tasks in the past few years and they can be grouped into two categories: single-stream cross-modal Transformers and two-stream cross-modal Transformers. Single-stream Transformers (Su et al., 2020; Chen

et al., 2020; Li et al., 2019; Huang et al., 2019) have only one encoder. The visual features and the linguistic features are concatenated together into a single input sequence. On the other hand, two-stream Transformers (Lu et al., 2019b; Yu et al., 2020; Tan and Bansal, 2019) have two independent encoders, one for the visual feature stream and the other one for the linguistic feature stream. Then a third encoder is used to capture the cross-modal relationship between the two modalities.

### 2.2 Visual-linguistic Pre-training

Visual-linguistic pre-training is widely applied to multimodal tasks using large-scale image captioning datasets, such as Conceptual Captions (Sharma et al., 2018) and MSCOCO (Lin et al., 2014). Two common pre-training tasks are *masked language modeling with visual clues* (MLM) and *masked RoI classification with linguistic clues* (MRC) (Su et al., 2020), which are the extensions of the original MLM task from BERT. *Word-region alignment* (Lu et al., 2019a; Chen et al., 2020), *image-text matching* (Chen et al., 2020), and *RoI feature regression* (Tan and Bansal, 2019) were also proposed. ERNIE-ViL (Yu et al., 2020) proposed the *scene graph prediction* task based on the semantic graphs parsed from the captions.

Other approaches for improving visual question answering performance were also proposed in addition to visual-linguistic pre-training. Wu et al. (2019) proposed to generate question-relevant captions jointly with answering the VQA questions. Kim and Bansal (2019) proposed to fuse the image, question, and answer inputs with an additional paragraph that provides a diverse and abstract description of the image. A similar idea is found in (Li et al., 2018) where generated captions are used to explain the image and combined with the question to produce more accurate answers. A detailed study (Singh et al., 2020) investigated the effect of the similarity between pre-training and fine-tuning datasets.

## 3 Our Method

### 3.1 Commonsense Inference Generation

Prior to our pre-training, we first generate commonsense inferences from the conventional image-caption pairs. In addition to the image domain and the caption domain, commonsense inferences are treated as a third knowledge domain that is required for our proposed pre-training. We take a visual-

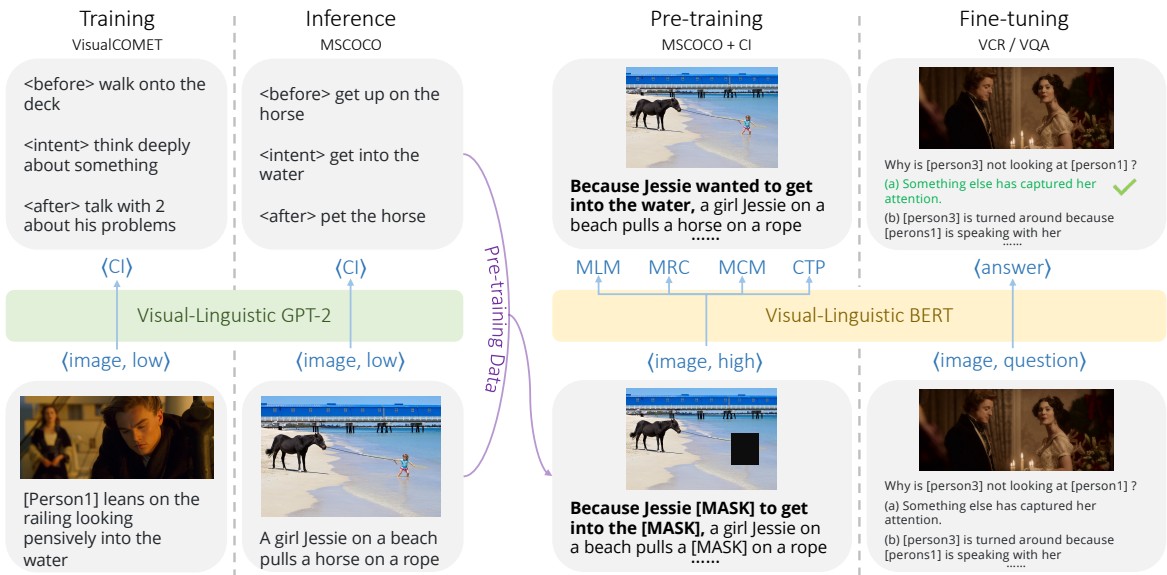

**Commonsense Inference Generation**

Training
VisualCOMET

Inference
MSCOCO

Commonsensical Training

Pre-training
MSCOCO + CI

Fine-tuning
VCR / VQA

Figure 3: An overview of our commonsensical pre-training framework. The left part shows the commonsense inference generator; the right part shows the pre-training and fine-tuning pipelines. The bold text in the pre-training stage is the generated commonsense inference (CI) and the template tokens. The blue arrows point from the inputs to the target outputs. That is, the bottom images and sentences are the inputs while the top images and sentences are the objectives. "Low" and "high" stand for low-level captions and high-leval captions, respectively.

linguistic GPT-2 as the commonsense generator and fine-tune it on the VisualCOMET (Park et al., 2020) dataset. VisualCOMET introduces three specific types of commonsense inferences given the images and the captions (termed as <event>), which are the possible incidents before or after the current event (<before>, <after>) and the potential intentions of the people in the image (<intent>). Different from the GPT-2 model proposed in VisualCOMET that requires additional location information, our GPT-2 only takes image and caption as inputs, as shown in the left half of Figure 3. In general, it can be easily applied to any existing large-scale image captioning dataset. In this paper, we generate commonsense inferences for the image-caption pairs in MSCOCO (Lin et al., 2014). Appendix A.3 includes more details about how our GPT-2 model is fine-tuned. Instead of simply concatenating the features from the three knowledge domains, captions and commonsense inferences are combined by a set of pre-defined templates. We term the combined sequence as the *high-level* caption. An example is shown in Table 1 and template details are included in Appendix A.4.

### 3.2 Commonsensical Pre-training

To exploit the additional knowledge inside the commonsense inferences, we introduce a novel com-

monsensical pre-training strategy, which consists of two new tasks: *masked commonsense modeling* (MCM) and *commonsense type prediction* (CTP). Both tasks are proposed to learn commonsense from a fine-grained and global aspect, alongside the conventional *masked language modeling with visual clues* (MLM) and *masked RoI classification with linguistic clues* (MRC). In MCM, instead of the random masking used in previous works (Su et al., 2020; Chen et al., 2020; Tan and Bansal, 2019; Devlin et al., 2018), we propose the *domain-wise adaptive masking* to adjust the masking ratio according to the semantic similarity between commonsense inferences and captions. We detail them one by one below.

**Masked Commonsense Modeling**   By incorporating commonsense inferences as the third knowledge domain additional to images and captions, we propose *masked commonsense modeling*. It is an extension of MLM with commonsense inferences as the input data and the *domain-wise adaptive masking* as the masking strategy. Each commonsense token is masked out by a probability that is controlled by the strategy detailed in the following "Domain-wise Adaptive Masking" subsection. The masked commonsense token $c_m$ is replaced with the special token [MASK]. The model aims to predict $c_m$ given the unmasked commonsense content

$c_{\backslash m}$ as well as the visual tokens $v$ and linguistic tokens $w$ by minimizing the negative log-likelihood:

$$\mathcal{L}_{\text{MCM}}(\theta) = -\mathbb{E}_{(c,w,v)\sim D} \log P_\theta(c_m | c_{\backslash m}, w, v)$$

where $\theta$ is the model parameters, and $D$ is the training dataset. We argue that the introduction of commonsense knowledge will help the model gain commonsensical reasoning ability.

For image regions and linguistic tokens, inheriting from previous works (Lu et al., 2019a; Su et al., 2020; Chen et al., 2020), we still use MLM and MRC tasks. One slight difference is that our MLM/MRC task is conditioned on both commonsense clues and visual/linguistic clues.

**Domain-wise Adaptive Masking** Since commonsense inferences are generated from low-level image-caption pairs by a commonsensical GPT-2, captions and commonsense inferences are likely to be semantically related to each other. It means that the model could potentially take the shortcut by excessively relying on the low-level captions when predicting the masked commonsense tokens and vice versa, which makes MLM and MCM easier to solve. Below is an example where [MASK]$_4$ is more likely be to predicted as "bridge" based on the linguistic clues of "overlooking the river" rather than visual clues, because "bridge" and "river" often coexist in a sentence:

> *"Before a man Casey in a wheelchair and another* [MASK]$_1$ *on a bench* [MASK]$_2$ [MASK]$_3$ *overlooking the river ,* **Casey needed to walk onto the** [MASK]$_4$."

To tackle this issue, we introduce the *domain-wise adaptive masking* strategy. In conventional settings, each linguistic token has a probability of 15% to be masked. Domain-wise adaptive masking considers the semantic distance between commonsense inferences and low-level captions and computes the masking ratio accordingly. It takes the sentence embeddings of commonsense inferences and low-level captions from a pre-trained BERT (Devlin et al., 2018) and calculates their cosine similarity. The similarity score is passed to a logistic function and rescaled to a high probability interval. We pick the rescaling interval $(0.5, 1.0)$ to ensure high masking ratio. A higher semantic similarity between the low-level caption and the commonsense inference leads to a higher masking

ratio on either the low-level captions or the commonsense inferences. Thus, the masking ratio is "adaptive" with respect to the embedding similarity. Detailed formulae and examples are shown in Appendix A.5.

During pre-training, adaptive masking is randomly applied on either low-level captions or commonsense inferences. Therefore, it is "domain-wise". When domain-wise adaptive masking is applied on low-level captions, it encourages the model to focus more on the visual clues for MCM. When domain-wise adaptive masking is applied on commonsense inferences, the same idea follows for MLM. The high masking ratio reduces the salience of one domain and elicits more advanced reasoning abilities, such as directly inferring commonsense knowledge from the images with only a few linguistic clues (heavily masked low-level captions).

**Commonsense Type Prediction** We also introduce a novel task of *commonsense type prediction* (CTP). It is an additional classification task that predicts the commonsense type (<intent>, <before> or <after>). Note that the template tokens are forced to be masked out in CTP since they are essentially the indicators of commonsense type. We also include the language modeling objective of these masked tokens in CTP. In general, it requires the model to perform commonsensical inference on the global relationship between commonsense and image-caption pairs.

## 4 Experiments

### 4.1 Implementation Details

GPT-2 is fine-tuned on VisualCOMET for 5 epochs using the AdamW optimizer with a learning rate of $5.0 \times 10^{-5}$. In pre-training and fine-tuning, we use the VL-BERT$_{\text{BASE}}$ configuration (Su et al., 2020), which is a single-stream cross-modal Transformer. VL-BERT is pre-trained for 10 epochs using the AdamW optimizer with a learning rate of $1.0 \times 10^{-7}$ and a weight decay of 0.0001. For downstream task evaluation on VCR, the pre-trained VL-BERT is fine-tuned for 20 epochs using the SGD optimizer with a learning rate of $7.0 \times 10^{-5}$ and a weight decay of 0.0001. For downstream task evaluation on VQA, the pre-trained VL-BERT is fine-tuned for 20 epochs using the AdamW optimizer with a learning rate of $6.25 \times 10^{-7}$ and a weight decay of 0.0001. Our experiments are conducted on 4 Nvidia TITAN RTX GPUs.

| Pre-training | VCR Q→A | VQA($v2$) | | |
|---|---|---|---|---|
| | | test-std | test-dev | val-human |
| None | 70.00 | 69.03 | 68.85 | 63.43 |
| Recognition-level | 70.46 (+0.46) | 69.95 (+0.92) | 69.71 (+0.86) | 66.09 (+2.66) |
| Commonsensical | **71.43** (+1.43) | **70.29** (+1.26) | **69.97** (+1.12) | **66.46** (+3.03) |

Table 2: Performance (accuracy) comparison on VCR and VQA among 3 settings: fine-tuning from scratch, fine-tuning from recognition-level pre-training, and fine-tuning from commonsensical pre-training. "Q→A" represents the question answering task from the validation set of VCR; "test-std" and "test-dev" represents the two testing phases of VQA; "val-human" represents the human-centric validation set of VQA.

## 4.2 Datasets

**Pre-training** We take MSCOCO (Lin et al., 2014) as our low-level image captioning dataset and apply our fine-tuned GPT-2 model on it to generate commonsense inferences. To avoid noisy labeling, we only augment the image-caption pairs which depict humans since it is counter-intuitive to infer intentions for non-human objects. Then commonsense inferences and low-level captions are combined by a set of pre-defined templates to form high-level captions.

**Fine-tuning** To evaluate the effectiveness of our commonsensical pre-training, we use Visual Commonsense Reasoning (VCR) (Zellers et al., 2019) and Visual Question Answer v2.0 (VQA$_{v2}$) (Goyal et al., 2017) for downstream task evaluation. The overall task of VCR is to select the correct answer (A) as well as the rationale (R) given an image-question (Q) pair. Existing works (Lu et al., 2019a; Su et al., 2020; Chen et al., 2020; Tan and Bansal, 2019; Yu et al., 2020) have shown that Q→A is a more challenging task, which is what we use to evaluate our proposed pre-training framework. VQA$_{v2}$ is another visual question answering task, where it primarily targets recognition-level understanding. In addition to the test set, we also evaluate our pre-training on a validation subset of VQA$_{v2}$, where only images that depict humans are considered. We term this subset as the human-centric VQA. We argue that these image-question pairs are more likely to be commonsensical (e.g., why is person...?). The subset is selected by the keyword matching of VQA's corresponding MSCOCO captions by a pre-defined human entity dictionary (e.g., student, firefighter).

## 4.3 Downstream Task Evaluation

To demonstrate the effectiveness of our pre-training framework, we fine-tune VL-BERT with different pre-train settings on VCR and VQA, including VL-BERT without pre-training, VL-BERT with conventional (i.e., recognition-level) pre-training, and VL-BERT with our commonsensical pre-training. Table 2 shows their performance comparison of accuracy on downstream tasks.

**VCR** The 1.43% performance increase on VCR from the no pre-training setting indicates the effectiveness of our proposed method and, in turn, the advantage of incorporating commonsense knowledge in pre-training. The slight 0.46% performance increase made by the conventional image-caption pre-training is consistent with the findings in VL-BERT and UNITER that the recognition-level pre-training might not be sufficient for commonsensical reasoning tasks. Our commonsensical pre-training enabled a 0.97% improvement over the recognition-level pre-training.

**VQA** As for VQA$_{v2}$, there is a 1.26% performance increase from no pre-training to our commonsensical pre-training in test-std set and a 1.12% increase in test-dev set. Our pre-training also improves over the conventional image-caption pre-training by 0.34% and 0.28%, respectively. Such increments are slightly smaller when compared to that on VCR. The reasone is that the questions in VQA mostly target recognition-level understanding (e.g., *What color is the ...?*, *What is the ...?*, *How many ...?*), the gap between recognition-level pre-training and fine-tuning on VQA is much smaller than that on VCR. In other words, commonsensical pre-training might be less necessary for VQA. On the other hand, the performance increment in the human-centric VQA is larger, at 0.37%. The comparison of no pre-training settings between "val-human" and the remaining VQA set (Table 2) has shown that human-centric VQA is a more challenging problem than the general VQA.

The performance gap between our results and the reported results from previous works (Su et al., 2020) is expected since our pre-training dataset is much smaller than the commonly used massive image-caption datasets, such as Conceptual Captions (Sharma et al., 2018). We also did not perform any hyperparameter tuning for the visual-linguistic BERT or fine-tuning of the image feature extractor Faster R-CNN, since we are aiming for rela-

| Pre-training | VCR Acc. (Q→A) |
|---|---|
| (a) None | 70.00 |
| (b) MLM$_{rec}$ | 70.46 |
| (c) MLM$_{rec}$ (Aug. + Rand-1 + DAM) | 70.55 |
| (d) MLM$_{rec}$ + MCM (Top-1) | 70.32 |
| (e) MLM$_{rec}$ + MCM (Rand-1) | 70.60 |
| (f) MLM$_{rec}$ + MCM (Rand-1 + DAM) | 71.02 |
| (g) MLM$_{rec}$ + MCM (Rand-1 + DAM) + CTP | 71.43 |

Table 3: Comparison of individual component of our proposed pre-training on VCR. MLM$_{rec}$: recognition-level pre-training tasks, including MLM and MRC; Top-1: pre-train using the top-1 commonsense inference from our fine-tuned GPT-2; Rand-1: pre-train using one commonsense inference randomly selected from the five candidates at each iteration; MCM: *masked commonsense modeling*; DAM: *domain-wise adaptive masking* strategy; CTP: *commonsense type prediction* task.

tive performance comparison rather than absolute improvement with respect to the state-of-the-art models.

### 4.4 Ablation Study

We further conduct a comprehensive ablation study to analyze the effect of each component in our commonsensical pre-training, as shown in Table 3. The ablation study is on VCR because we are more interested in commonsensical tasks and VCR is specifically designed for that.

The improvement from (d) to (e) indicates that the diversity of commonsense knowledge benefits the learning. When comparing (e) against (b), we can conclude that our commonsensical pre-training is indeed more advantageous than recognition-level pre-training. The performance increase from (e) to (f) demonstrates the effectiveness of domain-wise adaptive masking in encouraging better commonsensical multimodal learning by adaptively reducing the salience of one knowledge domain. The improvement of (g) over (f) demonstrates the effectiveness of the CTP task.

Since our high-level captions are essentially augmented captions with commonsense knowledge, we would like to see how it compares to other augmentation methods. One obvious baseline is to use a well-trained caption generator to obtain additional information for caption augmentation. We use OSCAR (Li et al., 2020), a state-of-the-art caption generator, to augment the original image caption with its generated recognition-level information. Then (c) represents the OSCAR-augmented recognition-level pre-training with Rand-1 and

| | Relevance (cap) | Relevance (img+cap) | Informa-tiveness | Diversity |
|---|---|---|---|---|
| Ground Truth | 3.88 | 3.95 | 3.29 | 3.21 |
| Generated | 3.43 | 3.48 | 3.58 | 3.66 |
| Ratio | 88.4% | 88.0% | 108.9% | 114.2% |

Table 4: Human evaluation of our generated commonsense inference on MSCOCO compared to the ground truth commonsense inference from VisualCOMET. "Ratio" is the score ratio of "generated" against "ground truth". The scores are on the scale of 0-5.

domain-wise adaptive masking applied. Although it improves from (b) approximately by 0.1%, it is much weaker than the increment between (b) and (f), at 0.56%. It demonstrates that the high-level commonsensical captions contain more useful and compatible information than the same amount of low-level captions do. Thus, we can conclude that the commonsense knowledge is indeed more compatible with the commonsensical reasoning ability for the downstream VCR task.

### 4.5 Commonsense Inference Evaluation

Because the MSCOCO dataset does not contain ground truth commonsense knowledge, we conduct a human evaluation on the quality of the commonsense inferences generated by our GPT-2. Following the evaluation method used in (Dua et al., 2021), we randomly sample image-caption pairs along with their corresponding generated commonsense inferences for MSCOCO and ground truth commonsense inferences from VisualCOMET, with a mixture ratio of 4:1.

We ask 10 human evaluators and have each of them evaluate 20 <image, caption, commonsense> entries without knowing whether the commonsense inferences are generated (MSCOCO) or annotated (VisualCOMET). Evaluators are asked to evaluate each commonsense inference from four dimensions on the scale of 0 to 5: *relevance (cap)*: how plausible is the commonsense inference provided the low-level caption only, *relevance (img+cap)*: how plausible is the commonsense inference given the image and the low-level caption, *informativeness*: how much extra information does the commonsense inference contain compared to the low-level caption, and *diversity*: the diversity of the five candidates commonsense inferences of each commonsense type.

We receive 12000 scores ($10 \times 20 \times 3 \times 5 \times 4$) in total. We then separate the results by generated (MSCOCO) versus annotated (VisualCOMET) and average the scores of each dimension. The results

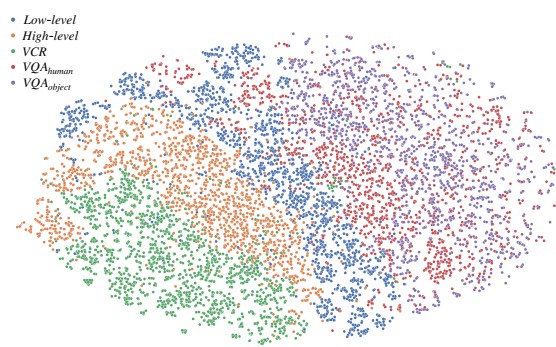

Figure 4: Corpus distribution of low-level captions, high-level captions, VCR, $VQA_{human}$, and $VQA_{object}$.

are shown in Table 4. The ground truth scores are treated as the reference for the quantified assessment of commonsense inferences quality. In terms of relevance measure, both caption-only and image-caption settings show considerable validity of our commonsense inferences on MSCOCO dataset, which is 88.4% and 88.0% of the ground truth relevance scores. It also shows that generated commonsense inferences are often more informative and diverse compared to the ground truth commonsense inferences. Detailed examples and analysis regarding the success and failure commonsense inference cases are included in Appendix A.6.

## 4.6 Qualitative Analysis

To understand how our proposed pre-training framework improves the downstream task performance, we perform a qualitative analysis regarding the semantic relationship among the conventional caption corpora, our pre-training corpora, and the corpora of VCR and VQA. We further separate VQA into $VQA_{human}$ and $VQA_{object}$, where $VQA_{human}$ is the human-centric VQA whose images depict human. We term $VQA_{object}$ as the object-centric VQA whose images depict things other than human. The visualization details are included in Appendix A.7. The distance between corpus distributions indicates different levels of information (e.g., recognition-level or commonsensical) and different knowledge domains (e.g., human-centric or object-centric) within each corpus.

It is easy to see that different datasets are well-separated in Figure 4. Considering the spatial relationship in the embedding space, the corpus distribution of VCR is the furthest away from that of $VQA_{object}$. This follows our intuition in that VCR and $VQA_{object}$ require different levels of understanding and reasoning and, additionally, VCR is

| Fine-tuning | $VCR_{sub}$ Acc. (Q→A) |
|---|---|
| VL-BERT | 68.30 |
| VL-BERT + Low-level | 70.87 |
| VL-BERT + High-level | 71.17 |

Table 5: Fine-tuning performance comparison with additional linguistic information (without, low-level, and high-level) on the VisualCOMET subset of VCR.

human-centric while $VQA_{object}$ is not. The overlap between $VQA_{human}$ and $VQA_{object}$ implies that a large portion of $VQA_{human}$ is still at recognition-level. The low-level pre-training dataset also contains human-centric captions, which explains the adjacency between low-level caption corpus and $VQA_{human}$. Although the low-level caption corpus is closer to VCR than VQA is to VCR, there still exists a gap between low-level caption corpus and VCR. Our commonsensical (i.e., high-level) pre-training corpus with commonsense inferences generated by GPT-2 successfully bridges the gap between the low-level caption corpus and the downstream commonsensical corpus, which explains part of the performance improvement by our proposed method. Additionally, the distance difference between high-level caption to $VQA_{object}$ and high-level caption to $VQA_{human}$ could explain why our proposed pre-training gains larger improvement on $VQA_{human}$. It demonstrates the pre-training can generalize better to tasks with similar knowledge domains, and implies that object-centric commonsense might be more suitable for improving $VQA_{object}$.

## 4.7 Fine-tuning with High-level Captions

Besides pre-training with high-level captions, we could also introduce low-level or high-level captions as additional information to support fine-tuning on VCR. We fine-tune the VL-BERT model on a subset of VCR where the images overlap with those in VisualCOMET (VisualCOMET uses a subset of VCR images, which takes up about half the size of the full VCR.). The three settings shown in Table 5 are the original fine-tuning of VL-BERT, fine-tuning with the addition of low-level captions, and fine-tuning with the addition of high-level captions. Results show that the high-level captions are also more useful than low-level captions in helping VL-BERT improve performance during the fine-tuning stage.

## 5 Discussion

**Summary** We propose a novel visual-linguistic pre-training framework that incorporates common-

sense knowledge in visual-linguistic pre-training to enhance the commonsensical reasoning ability of the model. The framework includes commonsense inference generation and two novel commonsensical pre-training tasks. The effectiveness of our pre-training framework is reflected through downstream task evaluation on VCR and VQA. We also perform extensive empirical analysis to get insights behind the improvement and demonstrate that our commonsensical pre-training is more compatible with commonsensical downstream tasks.

**Limitation**   It is noted that the current commonsensical pre-training is bounded by the performance of the commonsensical GPT-2. Theoretically speaking, this module is replaceable by any other visual-linguistic commonsense generators or retrievers. In addition, the scope of commonsense knowledge within this work only covers the temporal and intentional domains, while the potentials of utilizing other commonsense knowledge (e.g., object-centric) in pre-training remains unexplored.

**Future Work**   We plan to push the limits of the proposed pre-training framework by the following options: (1) Improve the quality of the existing commonsense generator; (2) Scale up the commonsensical pre-training with larger image-caption datasets, such as Conceptual Captions, and with larger vision-language models; (3) Explore more advanced commonsensical pre-training techniques other than using the extensions of the MLM objective. Another interesting direction would be exploring the pre-training effect of commonsense other than temporal and intentional knowledge.

**Acknowledgement**   This work was supported in part by DARPA MCS program under Cooperative Agreement N66001-19-2-4032. The views expressed are those of the authors and do not reflect the official policy of the Depart- ment of Defense or the U.S. Government.

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

## A   Appendix

### A.1   Transformer Revisit

The core component of Transformer (Vaswani et al., 2017) is Multi-head Self-Attention:

$$\text{MultiHead}(Q, K, V) = \text{Concat}(\text{head}_1, ..., \text{head}_h)W^O$$
$$\text{head}_i = \text{Attention}(QW_i^Q, KW_i^K, VW_i^V)$$
$$\text{Attention}(Q, K, V) = \text{softmax}(\frac{QK^T}{\sqrt{d_k}})V$$

where the trainable weights are $W_i^Q \in \mathbb{R}^{d_{\text{model}} \times d_k}$, $W_i^K \in \mathbb{R}^{d_{\text{model}} \times d_k}$, $W_i^V \in \mathbb{R}^{d_{\text{model}} \times d_v}$ and $W^O \in \mathbb{R}^{hd_v \times d_{\text{model}}}$; $d_{\text{model}}, d_k, d_v$ are hyperparameters and

$h$ is the number of self-attention heads. Because it is permutation equivariant, positional encodings are injected into the token embeddings.

BERT (Devlin et al., 2018) is a deep bidirectional Transformer, which is a stack of Transformer encoder layers:

$$X = \text{MultiHead}(E_{out}^{l-1}, E_{out}^{l-1}, E_{out}^{l-1})$$
$$X' = \text{LayerNorm}(X + E_{out}^{l-1})$$
$$E_{out}^l = \text{LayerNorm}(\text{FFN}(X') + X')$$

where $E_{out}^l$ are the encoder output at the $l^{th}$ layer. In BERT pre-training, *masked language modeling* (MLM) was proposed. It is a self-supervised setting where the model needs to predict the tokens that are masked out (with a probability of 15%) from the remaining tokens.

GPT-2 (Radford et al., 2019) is a multi-layer Transformer decoder where each decoder layer can be expressed as:

$$X = \text{MaskedMultiHead}(D_{out}^{l-1}, D_{out}^{l-1}, D_{out}^{l-1})$$
$$X' = \text{LayerNorm}(X + D_{out}^{l-1})$$
$$D_{out}^l = \text{LayerNorm}(\text{FFN}(X') + X')$$

where $D_{out}^l$ are the decoder output at the $l^{th}$ layer.

### A.2 VL-BERT Visual Features

Visual features and detected object boxes for both tasks are pre-computed and extracted by Faster R-CNN (Ren et al., 2015) that is pre-trained on the Visual-Genome (Krishna et al., 2016) dataset.

### A.3 Commonsense Inference GPT-2

The GPT-2 model of VisualCOMET relies on not only the low-level captions (named "event" in VisualCOMET) but also a "place" descriptor. In order to make the model more general, we fine-tune the GPT-2 model without the "place" information: it only takes as input a pair of image and low-level caption and generates commonsense inferences, as shown in the left half of Figure 3. The visual part of the GPT-2 model is unchanged, which depends on the visual features extracted by a Faster R-CNN model.

More specifically, the input sequence is `[<|b_img|>, ` $\mathbf{v}_0$ `, ..., ` $\mathbf{v}_m$ `, <|e_img|>, <|b_ev|>, ` $\mathbf{l}_0$ `, ..., ` $\mathbf{l}_n$ `, <|e_ev|>, <|before|>]`, where $\mathbf{v}$ and $\mathbf{l}$ are visual features and word embeddings, respectively; `<|b_···|>` and `<|e_···|>` are special tokens for marking the beginning and the end of the image and "event" sequences; the `<|before|>` token can also be replaced with `<|after|>` or `<|intent|>` to specify what type of commonsense inference to generate.

### A.4 High-level Caption Construction

After the three types of commonsense inferences are generated by GPT-2 for each image, we construct high-level captions by merging the original (low-level) caption with commonsense inference using the following templates:

- Before [low], [person] wanted to [commonsense inference].
- After [low], [person] will most likely [commonsense inference].
- Because [person] wanted to [commonsense inference], [low].

where [person] is the extracted subject name, [low] is the low-level caption and [commonsense inference] is the generated type-specific commonsense inference; all other tokens are named *template tokens* (e.g., Before ... wanted to). The "Inference section" of Figure 3 includes an example of such high-level caption.

We take the MSCOCO dataset (Lin et al., 2014) as our base pre-training dataset. It contains 533K unique image-caption pairs. Since VCR is a human-centric reasoning task, we filter MSCOCO by keyword matching with an pre-defined person-entity vocabulary (e.g., student, firefighter) and obtain its human-centric subset. We then generate human-centric commonsense inference on it. Our final pre-training dataset contains 257K unique low-level image-caption pairs and 3915K ($\approx 3 \times 5 \times 257$K) unique high-level image-caption pairs.

### A.5 Domain-wise Adaptive Masking Computation

The *domain-wise adaptive masking* ratio is computed by the equations below:

$$score = \text{cos\_sim}(\mathbf{h}_{low}, \mathbf{h}_{CI})$$
$$ratio = \text{Rescale}(\sigma(score))$$

where $\mathbf{h}_{low}$ is the sentence embedding for the low-level captions, and $\mathbf{h}_{CI}$ is the sentence embedding for its corresponding commonsense inferences. The sentence representation is the representation of the [CLS] token taken from BERT; $cos\_sim(\cdot)$ is the cosine similarity; $\sigma(\cdot)$ is the logistic function; Rescale is the min-max scaling, where the prior minimum and prior maximum are precomputed from the training data. In this work,

the posterior range is $(0.5, 1)$. Figure 5 is the histogram of the computed adaptive masking ratios from the training data with the mean ratio equals to 0.715. Examples of the calculated masking ratio are shown in Figure 6. Since "stop skiing" is more semantically related to "middle of a skiing jump", the function outputs a larger masking ratio compared to "fear for his life". The same idea follows as the "get served piazza" is more semantically related to "in front of two piazzas" compared to "gather the ingredients".

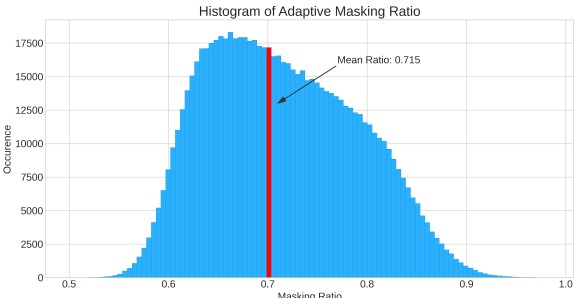

Figure 5: Histogram of the adaptive masking ratio from the training data.

## A.6 Commonsense Inference Evaluation

The generated commonsense inferences on MSCOCO are evaluated by human annotators from four dimensions on the scale of 0-5: relevant score given the caption only, relevant score given the image-caption pair, informative level, and diversity level. We include two examples in Figure 7, which corresponds to the success case and the failure case of the commonsense inference considering the evaluation scores. In the success case (Figure 7a), even though the caption mistakenly treats the Frisbee as a white ball, our commonsense inference GPT-2 successfully identifies the Frisbee and generates the commonsense inferences accordingly. The noisy caption explains the low scores in $rel_1$. The high $rel_2$ scores show the strength of our commonsense generator. Commonsense inferences in Figure 7b are evaluated as poorly generated. Both of its $rel_1$ and $rel_2$ scores are much lower. Compared to its image with the success case, we can see that it depicts a much larger scene where object details are harder to be perceived by the model. For example, the skier is doing tricks, while it can be ambiguous for the model to even identify human-alike shapes. However, the GPT-2 seems to recognize the scene as a big event. On the other hand, we can see that high information-level can be due to either in-

adequate captions, valid and informative commonsense inferences, or noisy commonsense inferences. The examples also show how the diversity-level can be positively correlated with the ambiguity-level of the images and negatively correlated with the relevant scores. It introduces some insights behind the higher informative and diversity score of the generated commonsense inferences in Table 4.

## A.7 Corpora Visualization

We randomly sample 10K "sentences" from each dataset to estimate their corpus distribution. For low-level pre-training and commonsensical pre-training, sentences simply refer to the low-level captions and high-level captions, respectively. For VQA, a sentence is the concatenation of a question and its corresponding ground truth answer with the highest confidence. The VQA corpus is further divided into human-centric VQA and object-oriented VQA. In VCR, a sentence is the concatenation of a question, its corresponding answer, and the ground truth rationale.

We use a pre-trained Sentence-BERT (Reimers and Gurevych, 2020) to retrieve the embedding of each sentence. Then each of the five datasets is represented by an embedding matrix of size $10,000 \times 768$, where 10,000 is the sample size and 768 is the hidden dimension size. We use the t-SNE nonlinear dimension reduction technique to project and plot the corpus distributions in a 2-dimensional space, as shown in Figure 4.

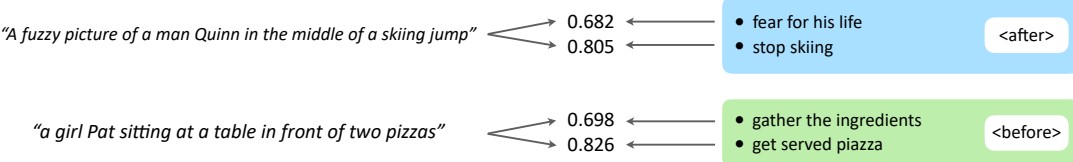

"A fuzzy picture of a man Quinn in the middle of a skiing jump"
0.682 ← • fear for his life
0.805 ← • stop skiing
<after>

"a girl Pat sitting at a table in front of two pizzas"
0.698 ← • gather the ingredients
0.826 ← • get served piazza
<before>

Figure 6: Examples of the calculated domain-wise adaptive masking ratio from low-level captions (left) and commonsense inferences (right).

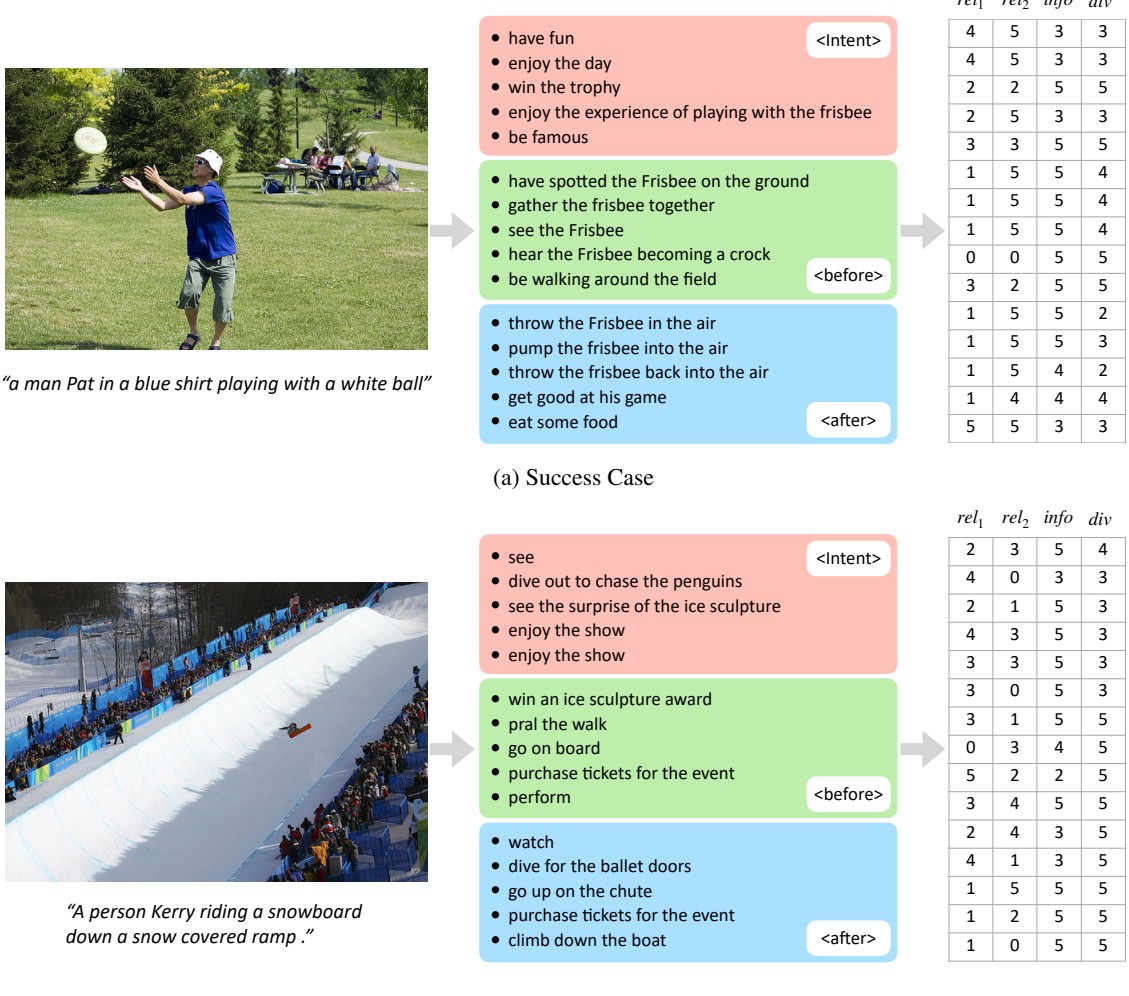

"a man Pat in a blue shirt playing with a white ball"

| | have fun | <Intent> | $rel_1$ | $rel_2$ | info | div |
|---|---|---|---|---|---|---|
| • | have fun | | 4 | 5 | 3 | 3 |
| • | enjoy the day | | 4 | 5 | 3 | 3 |
| • | win the trophy | | 2 | 2 | 5 | 5 |
| • | enjoy the experience of playing with the frisbee | | 2 | 5 | 3 | 3 |
| • | be famous | | 3 | 3 | 5 | 5 |
| • | have spotted the Frisbee on the ground | | 1 | 5 | 5 | 4 |
| • | gather the frisbee together | | 1 | 5 | 5 | 4 |
| • | see the Frisbee | | 1 | 5 | 5 | 4 |
| • | hear the Frisbee becoming a crock | | 0 | 0 | 5 | 5 |
| • | be walking around the field | <before> | 3 | 2 | 5 | 5 |
| • | throw the Frisbee in the air | | 1 | 5 | 5 | 2 |
| • | pump the frisbee into the air | | 1 | 5 | 5 | 3 |
| • | throw the frisbee back into the air | | 1 | 5 | 4 | 2 |
| • | get good at his game | | 1 | 4 | 4 | 4 |
| • | eat some food | <after> | 5 | 5 | 3 | 3 |

(a) Success Case

"A person Kerry riding a snowboard down a snow covered ramp ."

| | see | <Intent> | $rel_1$ | $rel_2$ | info | div |
|---|---|---|---|---|---|---|
| • | see | | 2 | 3 | 5 | 4 |
| • | dive out to chase the penguins | | 4 | 0 | 3 | 3 |
| • | see the surprise of the ice sculpture | | 2 | 1 | 5 | 3 |
| • | enjoy the show | | 4 | 3 | 5 | 3 |
| • | enjoy the show | | 3 | 3 | 5 | 3 |
| • | win an ice sculpture award | | 3 | 0 | 5 | 3 |
| • | pral the walk | | 3 | 1 | 5 | 5 |
| • | go on board | | 0 | 3 | 4 | 5 |
| • | purchase tickets for the event | | 5 | 2 | 2 | 5 |
| • | perform | <before> | 3 | 4 | 5 | 5 |
| • | watch | | 2 | 4 | 3 | 5 |
| • | dive for the ballet doors | | 4 | 1 | 3 | 5 |
| • | go up on the chute | | 1 | 5 | 5 | 5 |
| • | purchase tickets for the event | | 1 | 2 | 5 | 5 |
| • | climb down the boat | <after> | 1 | 0 | 5 | 5 |

(b) Failure Case

Figure 7: Examples of generated commonsense inference on MSCOCO with human evaluation. Left: image-caption pair as the inputs of the commonsense generator; Middle: generated commonsense inference; Right: human evaluation from four dimensions: $rel_1$ is the relevant score given the caption only; $rel_2$ is the relevant score given the image-caption pair; $info$ is the informative score; $div$ is the diversity score.