# OpenReview forum: "Bridging the Gap between Recognition-level Pre-training and Commonsensical Vision-language Tasks"
_aclweb.org/ACL/2022/Workshop/CSRR — ACL 2022 Workshop CSRR_

### Official Review · Reviewer_evpQ · 2022-03-23
**The paper proposes a pretraining loss to improve commonsense reasoning in visual-text models.**

**Rating:** 7
**Confidence:** 3

**Review:**

The main idea of this work is to use a model that generates commonsense phrases on image, caption pairs, and then pretraining another model on the images and text (original and generated) with some standard technique (e.g. MLM).
The authors present experiments on the VQA and VCR datasets, and show small improvements for these datasets.

The idea is clever, and it’s a nice use of existing dataset to improve commonsense reasoning to pretraining models.
The authors also present some analysis showcasing the benefits of their method


The paper is overall quite hard to read to my taste, it often gives details that focus on the low level, and doesn’t depict the high level, and on the other hand, there aren’t enough details to replicate this study.
In addition, the paper is somewhat overclaiming. The method for adding commonsense, is based on silver data (as it uses the output of another model), and is very limited - it only adds information about the timeline (whether the incident is before or after), and the possible intent. Not only these inferences can often be speculative and ungrounded, this is only a tiny bit of the commonsense space.
Another point to note is that the examples themselves (e.g. figures 1, 3) seem unnatural. This is fine if it eventually leads to improved performance, but it should be discussed in the paper.
Another point regarding the results is that these aren’t state-of-the-art results. This is fine on its own, but should definitely be mentioned and discussed.

The writing should be improved in terms of clarity and scrutiny, as well as the overclaiming (as discussed above). There are some issues with the citing style (\citet instead of \cite, e.g. in L.75, L.199).

---

### Official Review · Reviewer_Zykj · 2022-03-23
**Well-motivated novel pretraining method that shows some improvements but could benefit with some more setting clarifications/discussions.**

**Rating:** 7
**Confidence:** 3

**Review:**

This paper aims to improve downstream task performance on commonsense reasoning tasks with a pre-training scheme that involves both recognition and cognition tasks in an automated fashion (i.e., without needing more human annotation for the cognition part).

The paper treats the original MSCOCO [1] captions as “low-level” as they often involve solely recognizing objects/people/events. These low-level captions are augmented to high-level captions by using templates and a visual-linguistic GPT-2 model fine-tuned on VisualCOMET [2] without a <location> sense, but with the rest of the commonsense types such as <before>, <after>, and <intent>. **Therefore the 3 domains they consider in pre-training are: image, low-level captions, and high-level captions.**

The authors propose 3 new tasks/methods. The paper claims these methods improve pretraining incrementally (with VL-BERT) in addition to prior MLM & MRC schemes:

* **Masked commonsense modeling (MCM):**
    * The task of predicting a masked high-level caption domain token conditioned on the rest of the high-level tokens as well as the visual and low-level caption tokens.
    * For the other 2 domains, the authors still use MLM & MRC, conditioned on all 3 domains at the same time.

* **Domain-wise adaptive masking:** As the automatically generated high-level captions include parts of the low-level captions, the authors propose to adjust the masking ration between the two captions domains on their semantic similarity (cosine sim w/ BERT embeddings & some more processing).

* **Commonsense type prediction (CTP):** The task of predicting the type of the high-level commonsense captions by masking all of the template phrases.


### Strengths
**Background + motivation well-formulated:** The authors motivate their work well with prior research that separates recognition from cognition.

**Novelty:** They push forward that improvement in commonsense understanding has to start from pretraining rather than a fine-tuning scheme, which is a unique approach. They find novel methods to do so, by looking for ways to overcome specialization in a certain type of commonsense knowledge/ability.

**Ease of usage / scalability:** Their methods are automated and therefore don’t require more human annotation. Regardless of the automation, the results don't suffer form the basic templates.

**Ablation study:** The authors show the possible improvement role of each proposed method.


### Weaknesses
**Improvement margin:** It’s hard to tell from Table 2 whether the performance has improved vastly (~1% improvements). This gets harder with pre-training schemes combined with fine-tuning. It could be the case that a single point in accuracy improvement does matter. This is the case in many zero-shot learning tasks. However, in this approach the authors finetune on a separate set of the dataset tested. Therefore it’s hard to say whether it’s the finetuning that does most of the work and the pretrained embeddings are luckily well "initialized" with respect to the task or whether it actually improves the representations. I think that the human survey helps to back this to a certain extent, and that the method is interesting enough that a small margin doesn’t affect the rest of the paper. However, non-finetuned results on all methods (non-pretrained, recogniton, cognition) or some comments on why this small improvement matters would have been helpful.

**A little confused about the edge cases of masking even if it’s domain-wise:** It’s clear that MCM is conditioned on visual + low-level captions only, whereas MLM & MRC are conditioned on all three domains. But it’s a little hard to understand whether they are conditioned all independently or they could be all masked at the same time. For example it’s a little hard to understand from [lines 327-330] whether low-level captions and high-level inferences could be masked both at the same time, because of a combination of domain adaptive masking + MCM. It seems like no, and that’s the point of making sure a connection between images and the high-level captions exists, but it could be confusing to the reader. I think the paper could benefit from a clarification on this.

**Similarity of datasets:** It’s clearly stated in the paper that a subset of the VCR dataset is in the VisualCOMET dataset. It’s also stated that the high-level captions from VisualCOMET are mostly related to humans, while there is a "val-human" set separated from VQA. I wish there was a little more discussion on why this doesn’t or does affect the results, whether the improvement lies in picking the right datasets or not.

### Style
**The term commonsensical:** Although I understand that this term is used to separate cognition-level tasks from recognition ones, a reader from a commonsense research background in NLP may be confused (while originally this definition at [line 071] aims to reduce confusion). I would either stick to commonsense and say here we are solely referring to it from a cognitive points of view, or completely let it go and use words related to cognition. I understand that it’s hard to thread around what it might entail.

**Backbone citation:** I would change the citation at [line 045] to the actual GPT-2 citation and then move the visual-linguistic pretraining citation of [2] to the end of the sentence, as the rest of the sentence cites the actual backbone papers.

### References

[1] Lin, Tsung-Yi et al. “Microsoft COCO: Common Objects in Context.” ECCV (2014).

[2] Park, Jae Sung et al. “VisualCOMET: Reasoning About the Dynamic Context of a Still Image.” ECCV (2020).

---

### Official Review · Reviewer_qCsY · 2022-03-25
**Interesting idea, but results are not significant**

**Rating:** 6
**Confidence:** 5

**Review:**

Summary:

The authors propose a novel commonsensical vision-language pre-training framework as opposed to the current SOTA models which mainly focus on learning semantic connections between visual and language features/objects. The key novelty of the paper is to propose new pre-training tasks masked common sense modeling and commonsense type prediction. Extensive experiments on VCR and VQA demonstrate the effectiveness of the proposed approach.


Strengths:

The paper proposes an interesting way to enrich existing large language models with common-sense reasoning skills. This could be potentially adapted to other related tasks and domains.

I really liked the idea of getting commonsense inferences from Visual COMET and then using it in the newly designed pre-training tasks.

Although I am not fully convinced of the improvements shown in the paper, experiments sufficiently validate the conclusions drawn in the paper.


Weaknesses:

Here are my major concerns:

Authors showed improvements that are less than 2% in section 4.3. I worked with ViLBERT model before on both VCR and VQA and I found that any improvement < 2% cannot be considered as significant as it could simply be due to the variance in training. It would help (and is necessary) to present results with different seeds to validate the improvements.

For the VQA model, is Visual COMET really required? because most of the questions in VQA are not created to test the temporal aspect of the events. Maybe using the COMET (Jena Hwang et al., AAAI 2021) model might be a good choice here?

---

### Decision · Program_Chairs · 2022-03-28

Accept